# The Detection of Thoracolumbar Spine Injuries in Horses with Chronic Laminitis Using a Novel Clinical-Assessment Protocol and Ultrasonographic Examination

**DOI:** 10.3390/ani14091364

**Published:** 2024-04-30

**Authors:** Julia R. B. Guedes, Cynthia P. Vendruscolo, Paula K. A. Tokawa, Armando M. Carvalho, Philip J. Johnson, Rafael R. Faleiros

**Affiliations:** 1The EQUINOVA Research Group, Department of Veterinary Medicine and Surgery, Veterinary School, Federal University of Minas Gerais (UFMG), Belo Horizonte 31270-901, Brazil; armandodvm@gmail.com (A.M.C.); faleirosufmg@gmail.com (R.R.F.); 2Department of Veterinary Medicine and Zootechnics School (FMVZ), University of São Paulo (USP), São Paulo 13635-900, Brazil; cynthiaimpoluto@hotmail.com (C.P.V.); p.tokawa@gmail.com (P.K.A.T.); 3Equine Internal Medicine and Surgery, College of Veterinary Medicine, University of Missouri, Columbia, MO 65211, USA; johnsonpj@missouri.edu

**Keywords:** equine, laminitis, spine physical assessment, mobility tests, ultrasonography

## Abstract

**Simple Summary:**

Laminitis is a significant cause of horse mortality and it profoundly impacts the distal locomotor apparatus, causing intense pain primarily in the thoracic limbs. Affected horses exhibit a characteristic arched back and flexed thoracolumbar spinal posture. To the authors’ knowledge, there are no studies investigating the relationship between chronic laminitis and thoracolumbar spine injuries. The aim of this study was to investigate the occurrence and severity of injuries in the thoracolumbar spine of horses and the association with (*n* = 30) or without (*n* = 30) chronic laminitis. We developed a clinical-assessment method for the equine thoracolumbar spine, along with the equine Grimace Scale for pain assessment and ultrasound scans. The results indicated a significant association between chronic laminitis and thoracolumbar spinal injuries, with affected horses showing higher pain manifestations and increased injury rates based on clinical examination and ultrasound assessments. Horses with laminitis displayed a 14-times higher prevalence of relevant ultrasound injuries in the thoracolumbar spine. These findings emphasize the urgent need for further research to characterize and address these injuries, and an identification of targeted preventive and therapeutic measures to enhance the well-being of horses with chronic laminitis.

**Abstract:**

Postural adaptation is a prominent feature in horses affected by laminitis. Laminitis induces intense pain, especially in the forelimbs, prompting affected horses to assume a caudally displaced trunk posture, resulting in the hyperflexion of the thoracolumbar spine. This study assessed the nature and prevalence of thoracolumbar injuries in horses with chronic laminitis compared to horses without it. Sixty horses were used (thirty laminitic and thirty non-laminitic) of different athletic purposes and ages (2–20 years). The experimental protocol entailed a single assessment of horses’ thoracolumbar spines, utilizing physical examination by MACCTORE, a scoring system developed specifically for this study. Additional evaluations included the Grimace Equine Pain Scale (HGS) and ultrasound exams. Statistical tests were used to compare values (Mann–Whitney or *t*-test) and lesions prevalences (Fisher) between groups (*p* < 0.05). The results showed a higher pain manifestation (HGS and heart rate, *p* < 0.0001) and thoracolumbar-spine-injury levels in chronic laminitis horses, both in MACCTORE clinical examinations (11.7 ± 4.8 vs. 4.2 ± 3.3, *p* < 0.0001) and general ultrasonographic indices (39.6 ± 12.0 vs. 20.7 ± 7.1, *p* < 0.0001), including specific examination approaches for various spinal elements. Horses with laminitis presented with a 14-fold higher prevalence of ultrasound-relevant lesions in the thoracolumbar spine (CI: 4.4 to 50.6, *p* < 0.0001) compared to controls. These findings constitute new evidence of an association between chronic laminitis and the presence of thoracolumbar spine injuries in horses, which may be confirmed by more sophisticated study designs.

## 1. Introduction

Laminitis is a severe disease that affects the distal portion of equine limbs, and it is a significant cause of death in this species. Even so many years after the first reports of the disease, laminitis continues to be a challenge for veterinarians, farriers, and horsemen due to its complex pathophysiology, which can be related to several factors like sepsis, increased weight bearing, and endocrinopathic conditions (i.e., equine metabolic syndrome [EMS] and pituitary pars intermedia dysfunction [PPID]). This complexity arises because the hoof is a highly specialized integumentary portion responsible for supporting the weight of the horse and dissipating ground-reaction forces, where the connection of the hoof to the appendicular skeleton constitutes the suspensory apparatus of the distal phalanx [1,2]. Laminitis ensues when the structural and functional failure of this connection occurs, leading horses to having great difficulty in bearing weight on all four limbs and also moving, attributable to the pain they feel [3,4,5]. This disease can jeopardize not only the athletic career of horses, but primarily their well-being, which can even lead to death, generating significant financial and emotional costs for everyone involved in the process [6,7]. 

Traditionally, it has been suggested that laminitis affects the forelimbs more than the hindlimbs. According to a study by Leise et al. [8], the inflammatory process of laminitis occurs in the hindlimbs as much as in the forelimbs. However, the horse manifests greater signs of pain in the forelimbs because its center of gravity is located at the base of the withers. As a result, when the horse develops laminitis, it dislocates its trunk caudally, keeping the forelimbs extended cranially in an attempt to relieve pain in these limbs, as its body-mass center is shifted backwards and the weight on the forelimbs is decreased. With this antalgic presentation, the affected horse also develops an arched back and a flexed thoracolumbar spine, and the hindlimbs become displaced underneath the body, dramatically altering its anatomical posture and biomechanics [6].

In the last 20 years, there has been a significant improvement in the understanding of the physiological and pathological mechanisms that trigger the various forms of laminitis, which has allowed for earlier diagnosis and new resources for prevention [9]. Numerous biomechanical support methods and therapies focused on the digit have also been developed, allowing for considerable improvement in the health and well-being of horses with laminitis-related hoof structural failures [10]. However, it is known that there are repercussions beyond the hoof when a horse experiences laminitic pain. Despite all the scientific and technological advances, horses that live with the aftermath of laminitis still have important limitations in their athletic and work capacity. Some aspects remain obscure and demand research for an understanding of their comorbidities resulting from biomechanical changes caused by chronic laminitis. Among these changes, the one that draws the most attention is the abnormal posture that the horse adopts from the onset of pain, with the excessive and constant flexion of the thoracolumbar spine, causing a subsequent overloading of the muscle, ligamentous, articular, and bone structures, which can eventually be injured [9,11].

Even when a horse recovers from an acute laminitic episode and begins to improve, it may not always return to a normal posture, and may experience difficulties with locomotion, leading to an inability to resume an athletic life [12,13]. The reason for these longer-term difficulties is unclear; the possibility that spinal problems acquired as a result of the change in posture that the horse assumes during the period of chronic laminitis is deserving of further investigation [14]. Given this, this research delves into the hypothesis positing an association between thoracolumbar spine lesions and laminitis in horses. Employing a cross-sectional approach, this study aims to identify and compare the prevalence rates of these lesions in both clinically healthy and laminitis-affected horses. Furthermore, the investigation assessed the magnitude of these lesions and compared their clinical signs between the two groups.

## 2. Materials and Methods

### 2.1. Animals and Experimental Design

The experimental approach was approved by the Comissão de Ética no Uso de Animais (CEUA) (Ethical Commission of Animal Use) of Universidade Federal de Minas Gerais (UFMG), under protocol 230\2022. A cross-sectional design was used to investigate the association between chronic-laminitis consequences (exposure factor) with thoracolumbar pain/lesions occurrence (outcome). The studied population consisted of 60 horses (42 males and 18 females) aged between 2 and 20 years old and from different breeds. From those, 30 horses suffered from chronic laminitis and others were free of any limb disorders. 

The lamintic horses were randomly selected through contact with veterinarians and came from different breeding farms, and riding and training centers. Identifications of the horses’ ages, activities, and laminitis histories were provided by their caretakers. The inclusion criteria encompassed horses diagnosed and treated for laminitis within a timeframe of 3 to 12 months. These horses exhibited radiographic displacements of the third phalanx, as confirmed by X-ray examinations, and had no documented history of thoracolumbar pain or lesions before the onset of laminitis [15].

The non-laminitic horses were clinically healthy, without any documented history of thoracolumbar pain or lesions, and were randomly chosen from Haras Horizonte in Itapecerica, Minas Gerais State. They had no laminitis history or signs of laminitis, which was confirmed through gait examination and the use of hoof testers [16,17]. 

### 2.2. Physical Examination of the Thoracolumbar Spine

For the physical examination of the thoracolumbar spine, a protocol named the Clinical-Assessment Method of the Equine Thoracolumbar Spine (MACCTORE) was specifically developed, consisting of three stages: inspection, palpation, and mobilization. The method described herein was formulated to provide a thorough, systematic, and replicable clinical evaluation of the thoracolumbar spine. It was based on various protocols detailed in the international literature [11,17,18,19,20,21]. Preceding the current study, the examiner (JRBG) responsible for conducting the assessments applied this methodology across numerous horses, encompassing both clinically normal individuals and those experiencing acute or chronic back lesions. This training-phase prior use was supervised by another researcher (CV) and allowed for the establishment of typical responses within the studied population. During the study, all the assessments were conducted by this researcher (JRBG) in a calm, silent, and fly-free environment.

### 2.3. MACCTORE

The MACCTORE method consists of three assessment steps: inspection, palpation, and tests of mobility. Within each of these stages of the examination, there are predefined parameters that must be assessed. For each assessed parameter, a score was assigned from 0 to 3 (for inspection and palpation), or from 0 to 2 (for mobility tests), as detailed in Table 1. 

The assessment was carried out using the MACCTORE method, and its interpretation was performed after the researcher had been trained by an experienced professional, because while some horses respond to light digital stimulation, others need a more intense stimulation to show some reaction. After the training, all examinations were performed by the same researcher who had been through the process. 

#### 2.3.1. Inspection

The initial phase of the examination involved bringing the horses out of their stalls and cleaning their backs and loins using a brush, with potential washing for better visibility. Inspection is the first step of the physical examination protocol and was performed with the horse standing on all four limbs at a level surface. At this point, the horse’s conformation, the symmetry of muscle groups and/or their atrophy, the longitudinal alignment of the spine, its posture, the existence of abnormal curvatures in the spine such as lumbar kyphosis, thoracic lordosis, or scoliosis, sacral tuber asymmetry, or the presence of abdominal contractions were considered according to the method described by Fonseca [20].

The epaxial muscles that extend along the thoracolumbar region were also assessed through inspection. The atrophying of these muscle groups (with the prominence of the spinous processes) can indicate pain and/or injury in that region, according to Fonseca [22]. Moreover, the atrophying of these muscles can lead to a reduction of movement in painful areas and therefore should be considered a sign of thoracolumbar injury [19].

#### 2.3.2. Palpation 

Palpation of the thoracolumbar spine was performed while the horses were standing as squarely as possible. Palpation commenced with the first palpable thoracic vertebra (T4 or T5) and continued to the lumbosacral articulation, following the dorsal midline for the assessment of the supraspinous ligament and the dorsal alignment of the spinous processes [20,23]. The epaxial muscles were also palpated on both sides of the spinous processes, evaluating for changes in consistency, temperature, pain or fasciculations (Figure 1). Palpation was performed with the intent to identify the texture and consistency in the soft tissues and osseous structures, with attention to the supraspinous ligament, transverse processes, and paravertebral region for the assessment of epaxial muscles, assessing their symmetries, alignments, and presence of pain, as previously described [18,20,24].

#### 2.3.3. Mobility Tests 

Mobility tests involved passively moving the thoracolumbar spine through musculocutaneous stimulation to assess flexion range of motion and dorsal, ventral, and lateral extension. These tests, including dorsiflexion, ventral flexion, and lateral flexion with rotation, aimed to identify potential injuries in paravertebral soft tissues. Employing the MACCTORE system, each movement was scored based on normality, decreased or increased range of motion, and the presence or absence of muscle fasciculations, tail movement, limb reaction, vocalization, or evasion. Prolonged digital pressure at specific points along the spine was applied to evaluate muscle contraction, the horse’s attitude, tolerance to induced movement, range of motion, discomfort reaction, or any rejection of the examination.

##### Dorsiflexion and Ventral Flexion

The dorsiflexion and ventral-flexion tests were performed by means of an application of bilateral digital pressure in the form of tickling, symmetrically and simultaneously on both sides of the spine at specific points, while the horse was standing on the four limbs. The points defined and used in the MACCTORE method for flexion and extension of the spine were the following: (1) stimulation at the dorsal extent of T10, at the base of the withers, observing thoracic extension; (2) pressure with the tips of the fingers at the dorsal extent of T16, observing the thoracolumbar extension; (3) tickling at L4/L5, stimulating the lumbosacral extension; (4) pressure on the xiphoid cartilage, inducing a contraction of the *rectus abdominis* muscle and thoracolumbar ventral flexion; and (5) stimulation by tickling at the base\insertion of the tail, inducing lumbar and lumbosacral flexion in many cases (Figure 2 and Figure 3) [24,25].

##### Lateral Flexion and Rotation

This test was performed on both sides of the horse and consists of pulling, without aggressiveness, the base of the tail towards the side where the examiner is and applying pressure with the fingers against the body of the horse, at the height of the last ribs; thus, causing the horse to have a tendency to laterally flex its body and consequently the thoracolumbar spine (Figure 4) [24,26].

### 2.4. Facial Pain Score from Horse Grimace Scale 

The Horse Grimace Scale (HGS) was employed as an additional pain-assessment method to distinguish between the CON and CLG groups, addressing the subjective nature of clinical spine assessments. The objective was to provide a distinct and more objective measure of pain. The HGS protocol for facial pain assessment, as outlined by Dalla Costa et al. (2014) [27], was used to score all the horses in the experiment. Observations were conducted for five minutes immediately upon the horses’ arrival to the stable or pasture, before the commencement of the clinical examination. The same trained examiner evaluated all horses, and individual scores were recorded in their respective records.

### 2.5. Ultrasonography Examination of the Thoracolumbar Spine

The ultrasonographic equipment used was a portable veterinary unit (SonoScape X3V) with two transceivers (1–7 MHz convex probe and a 4–16 MHz linear probe). The target region was previously cleaned and prepared with 70% alcohol, and there was no need to clip/shave any horse. For the adequate measurement of the thoracolumbar spine, median and paramedian images were produced, the latter including images of the left and right sides along the entire thoracolumbar spine [3,20,28]. The ultrasonographic examination was performed in two steps: first, the cross-sectional assessment of the T17-L6 region using the convex probe (paramedian images), followed by the longitudinal assessment of the T5-L6 region using the linear probe (median images) (Figure 5).

In order to document ultrasonographic (US) findings, a specific US assessment record was developed with four specific targeted variables to be assessed in horses with chronic laminitis (i.e., horses with excessive and prolonged postures of flexion of the thoracolumbar spine) [29]. The selected variables included articular processes, epaxial muscle, spinous processes, and supraspinous ligament. The record contained a grading scale containing scores for each variable, as follows: articular process from 0 to 3; epaxial muscle from 0 to 4, with respect to the echogenicity of fibers, and from 0 to 2 with respect to muscle volume; the spinous process from 0 to 3, and the supraspinous ligament from 0 to 4. It included a legend with an explanation of what each score means. On the right side of the record, there was an individual score sum column for each variable and a space for the total US-score sum. This record was provided to the blind assessors along with the US images to be filled in.

The ultrasonographic assessments of all horses were performed by one singular veterinarian. However, the US images were blindly scored by two veterinarians that had at least a master’s degree. The assessment record was developed with the scores and the corresponding explanations of each determined variable, and was made available along with the US images. 

#### 2.5.1. Articular Processes and Epaxial Muscle

The articular processes of the dorsal intervertebral articulations were assessed in a cross-section, with the transducer located approximately 2 cm lateral to the dorsal midline of the spine. Two symmetrical images of the same intervertebral articulation, right side and left side, were obtained for a complete topographic view of the thoracic or lumbar vertebrae, and of the articulations, right and left (Figure 6 and Figure 7). Changes in the articular processes were classified as (0) normal or no irregularity; (1) slight irregularity of the articular surface; (2) moderate irregularity with a discontinuity of the articular surface; and (3) severe irregularity of the articular surface with a view of osteophytes in the articular line (Appendix A). The epaxial muscles assessed in this study included two main muscle groups: the multifidus muscle and the longissimus dorsi muscle. They were assessed through the same paramedian images and the same cross-section, produced by the convex probe, of those made for the assessment of the articular processes.

Muscle tissues were classified according to fiber echogenicity and muscle volume/tone. A score grading from 0 to 4 was established for echogenicity, where (0) represents normal echogenicity; (1) corresponds to a hypoechogenicity of fibers; (2) indicates an echogenicity of fibers; (3) corresponds to a hyperechogenicity of fibers; and (4) refers to heterogeneous fibers. In reference to muscle size, the classification was extended to the following scores: (0) normal; (1) muscle hypotrophy; and (2) muscle hypertrophy (Appendix A). 

#### 2.5.2. Spinous Processes and Supraspinous Ligament

The spinous processes and the supraspinous ligament were evaluated in longitudinal images obtained using the linear probe along the dorsal midline of the spine. The spinous processes were assessed regarding the regularity of the dorsal bone line, in relation to the bone surface and for the presence of bone fragments in the dorsal line. The scores established for the assessment of the spinous processes were the following (0) normality; (1) mild bone irregularity; (2) moderate bone irregularity with a possible loss of continuity of the bone line; and (4) severe bone deformity with the presence of osteophytes (Table 1).

The variations observed in the supraspinous ligament were characterized in relation to the echogenicity, in which the scores corresponded to the following: (0) normality of fibers; (1) observation of hypoechoic fibers; (2) observation of anechoic spaces between the fibers; (3) observation of hyperechogenicity between the fibers; and (4) observation of a heterogeneous environment (Appendix A) (Figure 8 and Figure 9).

### 2.6. Statistical Analysis of Data

The data were initially subjected to descriptive analysis to obtain the mean, standard deviation, standard error, median, and quartiles of each variable. Subsequently, the values of the laminitic and non-laminitic horses were compared in each variable using the unpaired Student’s *t*-test for normally distributed (parametric) data (based on the results of Shapiro–Wilk and Kolmogorov–Smirnov tests) and common variance (F-test). The Mann–Whitney test was used for nonparametric data. In addition, the Spearman test was used to correlate grades between researchers and MACCTORE and ultrasonographic data.

In order to detect the association between exposure to chronic laminitis and thoracolumbar pain/lesion outcomes, the prevalence of back pain and US injuries was compared between laminitic and non-laminitic horses using the chi-square test. Furthermore, the frequency distribution of horses affected with scores of clinical and severe US injuries, whose cutoff point corresponded to the 75th percentile (P75) of the entire population in each variable, were equally compared. If the calculated *p* < 0.05, the null hypothesis of independence was rejected, and a significant association between the two attributes was detected. The strength and relevance of statistically significant associations between exposure and specific outcome parameters were subsequently evaluated through the calculation of the Cramer’s V coefficient [30] and the calculation of the prevalence ratio [31]. 

The data were analyzed using statistical software (GraphPad Prism 10), considering a significance level of *p* < 0.05 for all tests. The Cramer’s V coefficient (scale 0 to 1) was obtained as previously described [31], and the association between the exposure factor and outcome was classified as negligible (up to 0.10), weak (0.11 to 0.20), moderate (0.21 to 0.40), marked (0.41 to 0.60), strong (0.61 to 0.80), and intense (0.81 to 1.00). 

## 3. Results 

### 3.1. Clinical and US Assessment

There was no difference in body-weight values between the groups; however, the laminitic horses showed an average height about 5% greater than the others (*p* < 0.0001). Horses with chronic laminitis had a significantly (*p* < 0.0001) higher heart rate (HR) and lower rectal temperature (RT) than non-laminitic horses (Figure 10). Increased scores (*p* < 0.0001) were also found in laminitic horses for the MACCTORE score (2.7-fold) and in the face pain scale (28-fold) compared with the non-laminitic horses (Figure 10).

All ultrasonographic parameters were observed to be significantly higher (*p* < 0.0001) in laminitic horses (Figure 11). Moreover, a significant positive correlation (*p* < 0.0001) between the two independent evaluators was detected for the articular (r = 0.85) and spinous (r = 0.61) processes, the supraspinous ligament (r = 0.66), the epaxial muscles (0.31), and the total ultrasound assessment (US Total), which was the sum of all assessments (r = 0.83). A significant positive correlation (r = 0.56, *p* < 0.0001) was also found between the US total and MACCTORE scores (Figure 12).

### 3.2. Relative Prevalence of Ultrasound Changes in the Equine Thoracolumbar Spine 

In this first prevalence analysis, the relative prevalence rates of ultrasound injuries considering all the grades and types of lesion were detailed by the anatomical structure of each vertebra, antimere, and group (Figure 13, Figure 14, Figure 15 and Figure 16). In this analysis, the prevalence of lesions in the articular processes (vertebrae L1, L2, L4 and L5) and in the supraspinous ligament (vertebrae T13, T15, T16, and T17) were higher in the laminitic horses compared to the others (*p* < 0.05). 

### 3.3. Prevalence of Horses Affected by Severe Clinical and Ultrasound Injuries

This second analysis was focused on comparing the prevalence of horses affected by severe injuries (i.e., those with higher clinical significance) independently of the site (vertebrae and antimere) between the two groups (Figure 17). The cutoff point corresponded to the 75th percentile (P75) of the entire population in each variable. On average, horses with laminitis had a markedly higher prevalence of severe injuries compared to the control group in the articular processes (330%), spinous processes (700%), supraspinous ligament (500%), epaxial muscles (600%), and in the sum of ultrasound scores (1400%). Regarding the MACCTORE clinical score, only horses with laminitis presented severe injuries. According to the Cramer’s V coefficient, the association between chronic laminitis and severe thoracolumbar US lesions was classified as marked (spinous and spinous processes), strong (supraspinous ligament and epaxial muscle), and intense when considering the US total score. The association between chronic laminitis and severe clinical manifestations of thoracolumbar lesions was at the upper limit of the moderate classification (Cramer’s V = 0.39).

## 4. Discussion

Although its small number of subjects and the inability to establish causality (its cross-sectional design), this study provides the first piece of scientific evidence that horses with chronic laminitis are more affected by thoracolumbar spine injuries than non-laminitic ones. It is well known that this kind of design cannot be used to determine the causes of disease because the temporality is not known. In other words, this study was not designed to prove that chronic laminitis sequels are causing back pain or the opposite. However, the statistical association between the previous condition (chronic laminitis) and the studied health effect (back pain and lesions) could be established based on the statistically higher prevalence rates of thoracolumbar spine injuries in laminitic horses.

The two groups of horses were not genetically and phenotypically identical. However, the low magnitude of the height difference (5%) and the absence of difference in body weight between the groups demonstrates reasonable uniformity between them. This discrepancy may be explained by the predominance of Quarter Horses in the non-laminitic group, which tend to be robust animals but of a smaller stature. It is also important to notice that the laminitic group includes horses with all types of laminitis, including horses with laminitis associated with sepsis/SIRS, and endocrine disorders. Therefore, horses varied in the spectrum of how much they weighed, from those who were very thin and debilitated and unable to stand up, to obese horses with fat accumulation in the crest of the neck and the base of the tail, for example, with signs of EMS [32].

Horses with chronic laminitis had a 28% higher HR than the non-lamintic horses. Elevated HR has been previously attributed to pain associated with laminitis [33,34]. This relevant clinical datum found in the experiment reinforces what the literature has already documented well regarding the relationship between physical parameters and clinical signs of laminitis. Due to the degree of pain that horses feel, from the acute phase of laminitis to the chronic process of the disease, the HR of these horses tends to have higher values than clinically healthy horses [35,36]. 

In contrast to the HR, which tends to be higher during both the acute and chronic phases of laminitis, the respiratory rate (RR) normally only increases in the initial phase of the disease and then tends to stabilize at normal physiological values [4,37]. The result regarding RR in this experiment is consistent with what has been reported in the literature: no statistical significance was found in RR between the non-laminitic and lamintic groups. Furthermore, reinforcing this result, the lung auscultation of all horses examined for this experiment was normal, without any abnormal sounds or pulmonary crackles, showing the absence of any pulmonary alterations that could interfere in the RR.

The rectal temperature in the non-laminitic group was slightly higher (but remained within the normal physiological range for the equine species) [20]. Meanwhile, this finding indicates that the disease process was actually chronic and equally important, and that the horses in this group did not present any other acute inflammatory process or any other disease that caused feverishness at the time at which they were assessed for the experiment. 

The mean MACCTORE score of the lamintic group was 277% higher than the score of the non-laminitic group. This scale was specifically developed for the physical evaluation of the thoracolumbar spines of horses, and evaluation sites were selected based on previous studies [11,20]. Other studies have already sought to develop methodologies for the physical assessment of the equine spine, with the purpose of studying injuries, searching for causes of and contributions to lameness, and also to pain in the muscles and joints of the lower back. An experiment conducted by Mayaki et al. [38] showed that painful responses to palpation of the back, muscular hypertonicity of the back, and stiffness of the thoracolumbar articulation are useful in determining pain in the lower back in horses. However, Fonseca et al. (2005) [22] and Fonseca et al. (2011) [24] established a coherent methodology for the physical assessment of the equine thoracolumbar spine and legitimized that when it is performed well, it can be very effective in diagnosing and monitoring pain and injuries in equine thoracolumbar spines. 

In the present study, the Horse Grimace Scale (HGS), validated by Della Costa et al. [8], was used for the purpose of having another variable to characterize painful stimulation present in the study that could help validate the MACCTORE method [7,39]. Considering the HGS was 38-times higher in the laminitic horses, this result confirmed what the literature had already been reporting about the painful symptoms of the disease under study, and corroborated the findings of the MACCTORE score.

The results obtained through the ultrasonographic examination were very enlightening regarding the identification of anatomical segments involved in the clinical signs detected by the MACCTORE method. All the examined structures from laminitic horses had increased grades for lesions compared with normal horses. Of course, it is impossible to separate lesions that developed before the onset of the laminitis from those potentially related to this disease. Also, the epidemiological design here employed did not allow us to make any statement about the laminitis process or its posture-induced changes being the real cause of such increases. However, considering the biomechanics of the equine thoracolumbar spine, these findings could be expected. For example, the hyper-flexed posture assumed for laminitic horses for long periods [40,41] could be related to greater friction in the articular processes and greater tension in the supraspinous ligament. Thoracolumbar hyperflexion related to other causes has been suggested to cause tissue reactions and injuries in these structures [26,42]. Additionally, similar increased lesions in the spinous processes could share the same mechanism, associated with the supraspinous ligament, due to the intimate anatomical relationship that they have. 

The most frequently observed changes in the epaxial muscles in horses with chronic laminitis were changes in the echogenicity of the muscle fibers in ultrasound assessments. The injury that was most seen in ultrasounds throughout the entire length of the epaxial muscles of the thoracolumbar spine in these horses, from T17 to L6, was a heterogeneous appearance followed by an increase in the echogenicity of the muscle fibers. Even though muscle volume was not used as a technique in this experiment, Oliveira et al. [2] showed that the ultrasound biometry of the multifidus muscle at the height of L5 is efficient in measuring the hypertrophy of this equine muscle group when training with specific reins that stimulate thoracolumbar flexion, especially in the lumbosacral region, during exercise. This reference and some others [37,43] corroborate the understanding that training programs or continuous work/effort, by modifying the horse’s posture in motion and causing biomechanical alterations, can induce functional adaptations and modifications in the size of certain muscles. Therefore, the continuous muscle effort due to the constant and intense flexion of the thoracolumbar spine could be a plausible explanation for increased scores for epaxial muscle lesions in the laminitic group. 

It is worth considering that the EM variable was the one that had the lowest correlation between the two assessors. This may be due to the difficulty in assessing muscle fibers through ultrasounds, regarding their echogenicity, or otherwise due to the fact of measuring muscle groups by biometry, because of the extent of the thoracolumbar fragment assessed in this study.

The total ultrasound assessment (total US) was a sum of all assessments, and their respective scores, of each structure outlined above. The significant and positive correlation between this variable and MACCTORE grades is further evidence that this proposed scale has the potential to be used for experimental and clinical assessments of the equine thoracolumbar spine in future studies. 

Looking at the prevalence of ultrasound injuries, according to the anatomical structure, examined vertebra, antimere, and group (Figure 13, Figure 14, Figure 15 and Figure 16), it was interesting to observe that the prevalence of injuries with higher AP scores were more usual in L1, L2, L4, and L5, and that lesions in the spinous processes were more usual between T11 and L2 in both groups, with no difference between them. This may be justified by the biomechanical functioning of the thoracolumbar spine. The greatest reactions in the SP would be expected as a consequence of extension movements of the spine, which would cause them to touch and rub against each other [25]. However, the opposite occurs in horses with chronic laminitis, who are in an antalgic posture, and, therefore, in a flexion movement of the thoracolumbar spine, but we still see changes in the SP, possibly due to the tension of the SSL, which is in close anatomical contact with the processes, and also due to the athletic function/activity that these horses have already previously performed before the disease [2,27].

Regarding the prevalence of lesions in the supraspinous ligament (SSL), there was a significant difference between the groups. The most severe SSL lesions were found mostly between T13-T17 in the CLG group, supporting an earlier assumption that horses with chronic laminitis (in an antalgic position with a hyperflexion of the thoracolumbar spine) can overload the SSL due to axial biomechanics [13]. 

The comparison of the frequency of severe injuries in the thoracolumbar spine between the two groups emerged as the pivotal analysis in this study. This analysis proved not only the association between laminitis and thoracolumbar lesions, but also underscored the robustness of this relationship according to the Cramer’s V coefficient. The prevalence rates demonstrated that horses with laminitis have a significantly higher likelihood than clinically healthy horses of experiencing severe injuries in various anatomical structures, including articular processes (3.3 times), spinous processes (7 times), the supraspinous ligament (5 times), epaxial muscles (6 times), and the entire thoracolumbar segment (US total, 14 times). Concerning the MACCTORE clinical score, only horses with laminitis exhibited severe injuries, indicating mathematically that the probability of having a severe MACCTORE grade is infinitely more pronounced in laminitic horses.

While acknowledging the significance of the findings in this study, it is crucial to recognize its methodological limitations. This is a primary study specifically designed to verify if the hypothesized association does indeed exist, which was conducted by a single research group using novel methods, thus necessitating a further scientific validation of the causality relationship between laminitis and thoracolumbar injuries through other controlled or cohort studies. Additionally, the size of the studied population sample might be deemed small for an epidemiological study, and the non-laminitic group could be more effectively represented to ensure greater diversity in the population under examination and to better establish an association without many potential confounders and co-factors, which could have influenced this first trial. 

New efforts are urgently needed to expand the characterization of the injuries demonstrated here and to propose specific preventive and therapeutic measures for the spine that provide greater effectiveness in the rehabilitation and well-being of equines with chronic laminitis.

## 5. Conclusions

Although preliminary, the results are sufficient to demonstrate a significant association between chronic laminitis and the occurrence and intensity of injuries in the equine thoracolumbar spine. New studies using more sophisticated designs are warranted to better delineate this association and to understand the clinical relevance of such findings. Additionally, the MACCTORE method was successfully employed in all horses and showed consistency and significant correlation with the ultrasonographic evaluation for detecting and scoring the thoracolumbar spines of horses with or without chronic laminitis.

## Figures and Tables

**Figure 1 animals-14-01364-f001:**
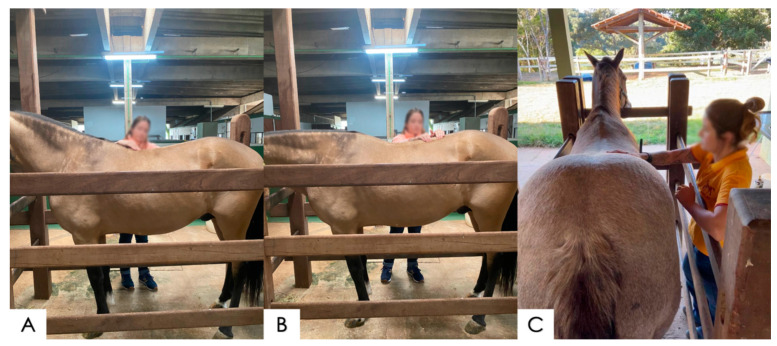
Palpation of the equine thoracolumbar spine. (**A**) Palpation of the spine along its dorsal midline for the assessment of the supraspinous ligament and the dorsal alignment of spinous processes; (**B**) palpation in the paravertebral line on the left side of the horse’s spine for a better evaluation of the supraspinous ligament and an assessment of the pain sensitivity, and (**C**) palpation of the longissimus dorsi and lateral iliocostalis muscles (right side of the horse) to the spinal midline of the horse.

**Figure 2 animals-14-01364-f002:**
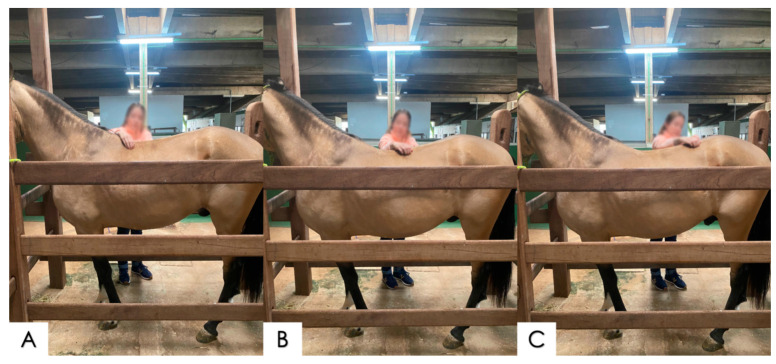
Digital stimulation points for thoracolumbar spinal extension. (**A**) Stimulation at the dorsal extent of T10, at the base of the withers, observing thoracic extension; (**B**) stimulation with the tips of the fingers of the dorsal extent of T16, observing thoracolumbar extension; and (**C**) tickling at L4\L5, stimulating lumbosacral extension.

**Figure 3 animals-14-01364-f003:**
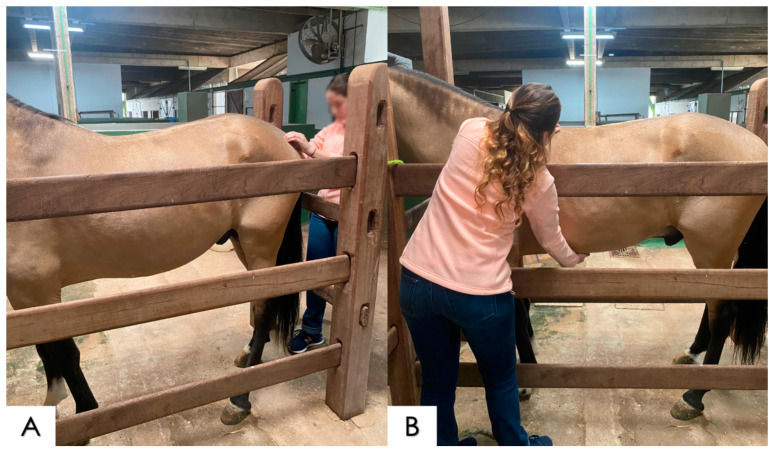
Digital stimulation points for equine spine flexion. (**A**) At the base of the tail, tickling with the tips of the fingers or light digital pressure to induce lumbar and lumbosacral flexion; (**B**) caudally to the xiphoid cartilage, tickling stimulation to induce thoracic flexion.

**Figure 4 animals-14-01364-f004:**
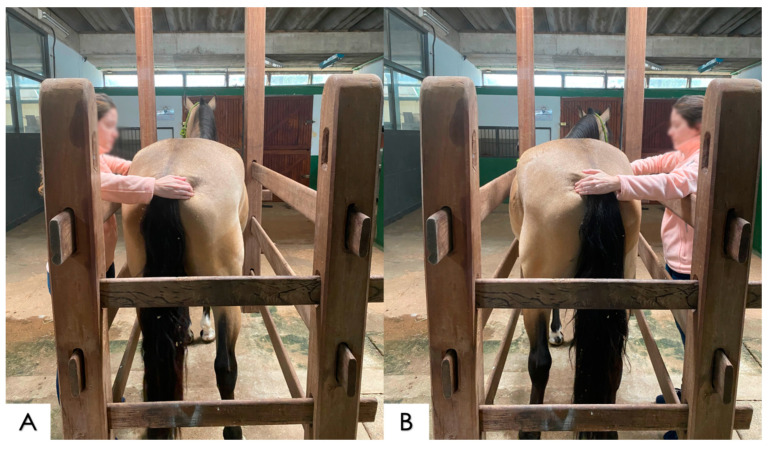
Examination of the spinal lateral-flexion test. (**A**) Application of a digital stimulation at the level of left rib #17, directing pressure in the opposite direction to the examiner, positioned on the left side of the horse, and at the same time, traction at the base of the tail towards the examiner to obtain a lateral flexion of the spine to the left side; (**B**) application of a digital stimulation to the level of right rib #17, directing pressure in the opposite direction to the examiner, positioned on the right side of the horse, and at the same time, traction at the base of the tail towards the examiner to obtain a lateral flexion of the spine to the right side.

**Figure 5 animals-14-01364-f005:**
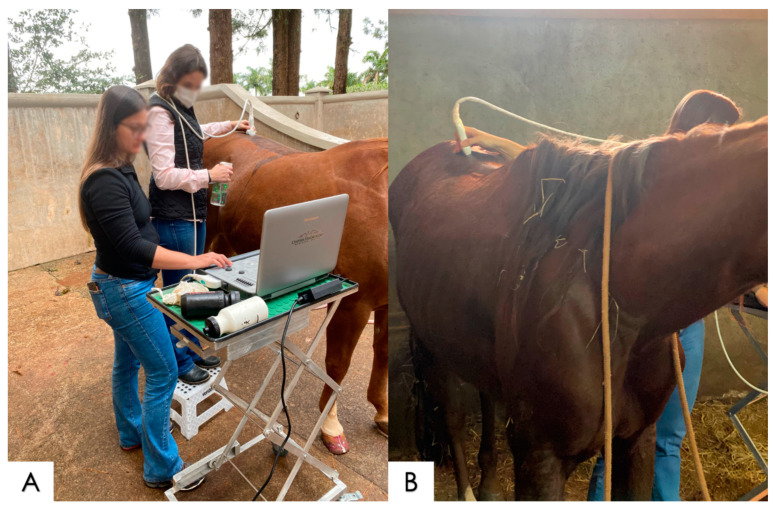
Ultrasonographic examination of the thoracolumbar area performed during the present study. (**A**) The cross-section assessment that was performed from T17 to L6, on both sides, using the convex probe, producing paramedian images, and (**B**) the longitudinal assessment that was performed from T5 to L6, on the dorsal midline of the spine, with the linear probe producing median images.

**Figure 6 animals-14-01364-f006:**
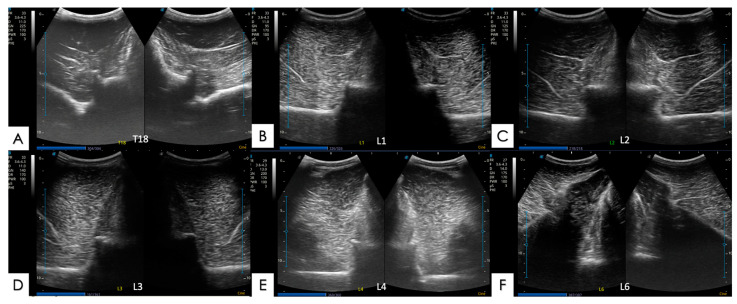
Ultrasound images with no abnormalities in cross-sectional, symmetrical views (right and left) of different equine thoracolumbar vertebrae taken during the current study. (**A**) Vertebra T18; (**B**) vertebra L1; (**C**) vertebra L2; (**D**) vertebra L3; (**E**) vertebra L4; and (**F**) vertebra L6.

**Figure 7 animals-14-01364-f007:**
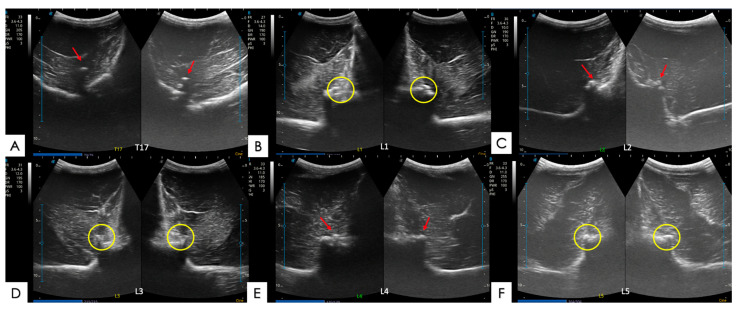
Ultrasound images with abnormalities in symmetrical cross-sections (right and left sides) of different thoracolumbar equine vertebrae, made during the current experiment. (**A**) T17 vertebra with osteoarthritis and the presence of osteophytes close to the articular processes on both the left and right sides (red arrows); (**B**) L1 vertebra with osteoarthritis with a moderate to severe loss of articular regularity on both sides (yellow circles); (**C**) L2 vertebra with osteoarthritis, a mild loss of articular regularity, and the presence of osteophytes on both sides (red arrows); (**D**) L3 vertebra with osteoarthritis with a moderate loss of articular regularity on both sides (yellow circles); (**E**) L4 vertebra with osteoarthritis with a mild articular irregularity of the left side (red arrows); and (**F**) L5 vertebra with osteoarthritis with mild articular irregularity and the presence of osteophytes on the left side (yellow circles).

**Figure 8 animals-14-01364-f008:**
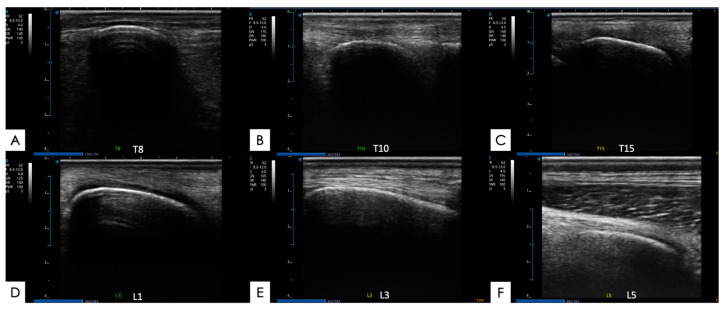
Ultrasound images with no abnormalities, in longitudinal sections, of different thoracolumbar equine spine vertebrae, taken during the current experiment. (**A**) SP of T8 vertebra; (**B**) SP of T10 vertebra; (**C**) SP of T15 vertebra; (**D**) SP of L1 vertebra; (**E**) SP of L3 vertebra; and (**F**) SP of L5 vertebra.

**Figure 9 animals-14-01364-f009:**
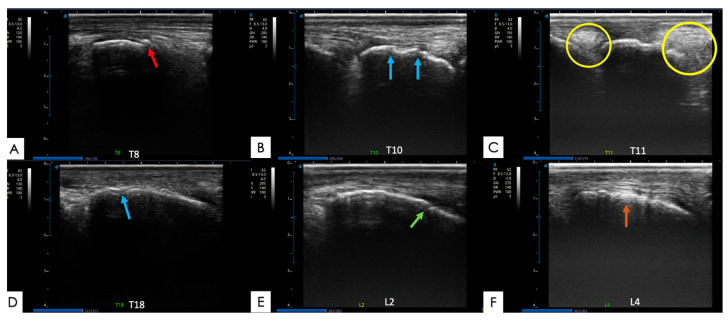
Ultrasound images with abnormalities, in longitudinal sections, of different thoracolumbar equine spinal vertebrae. (**A**) SP of T8 vertebra with an osteophyte presence (red arrow); (**B**) SP of T10 vertebra with moderate to severe irregularity (blue arrows); (**C**) SP of T11 vertebra with a moderately irregular surface and SSL with circumscribed hyperechoic areas (yellow circles); (**D**) SP of T18 vertebra with mild irregularity and remodeling (blue arrow); (**E**) SP of L2 vertebra with a moderately irregular surface and a loss of continuity (green arrow); and (**F**) SP of L4 vertebra with moderate to severe irregularity and SSL with presence of hyperechoic fibers (orange arrow).

**Figure 10 animals-14-01364-f010:**
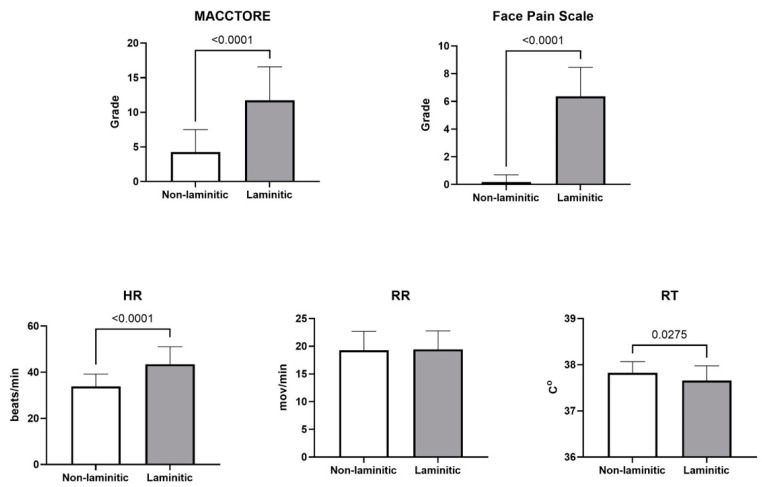
Means and standard deviations for MACCTORE, pain face scale, respiratory (RR) and cardiac (HR) rates, and rectal temperature in horses with or without chronic laminitis.

**Figure 11 animals-14-01364-f011:**
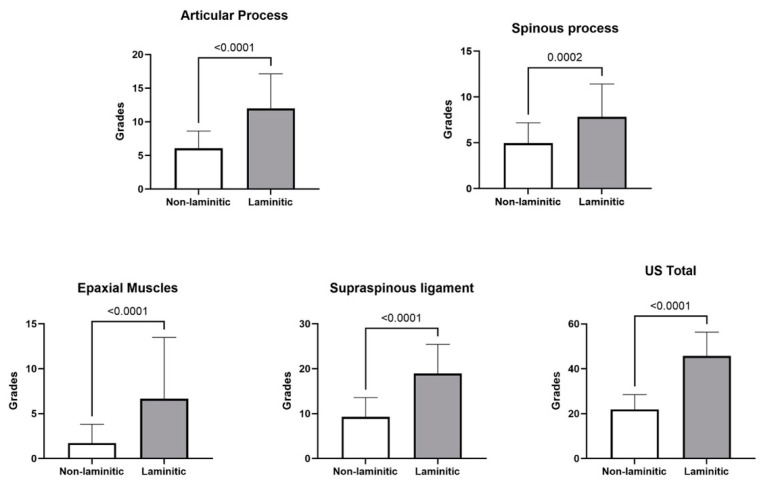
Mean and standard deviation grades for lesions detected by ultrasonographic assessment in the articular and spinous processes, the supraspinous ligaments, the epaxial muscles, and the sum of all grades (US total) in horses with or without chronic laminitis. *p* values are represented in the lines over bars.

**Figure 12 animals-14-01364-f012:**
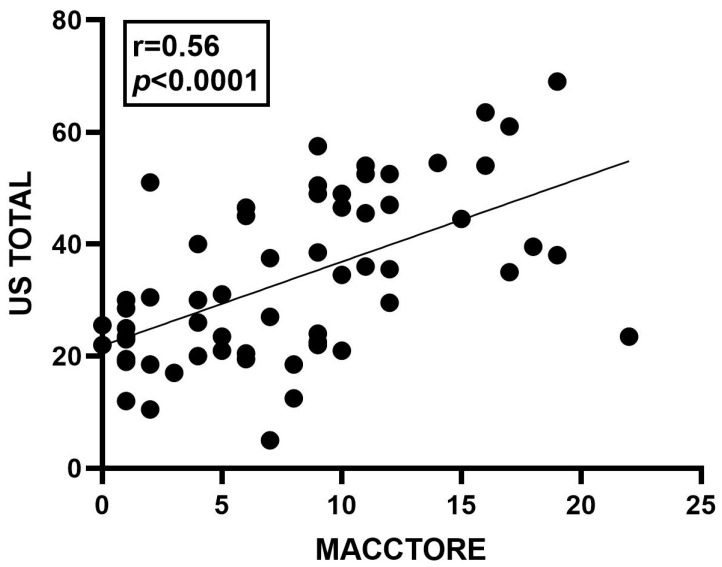
Spearman correlation between total ultrasound assessment (US total) and the MACCTORE grades in horses with or without chronic laminitis.

**Figure 13 animals-14-01364-f013:**
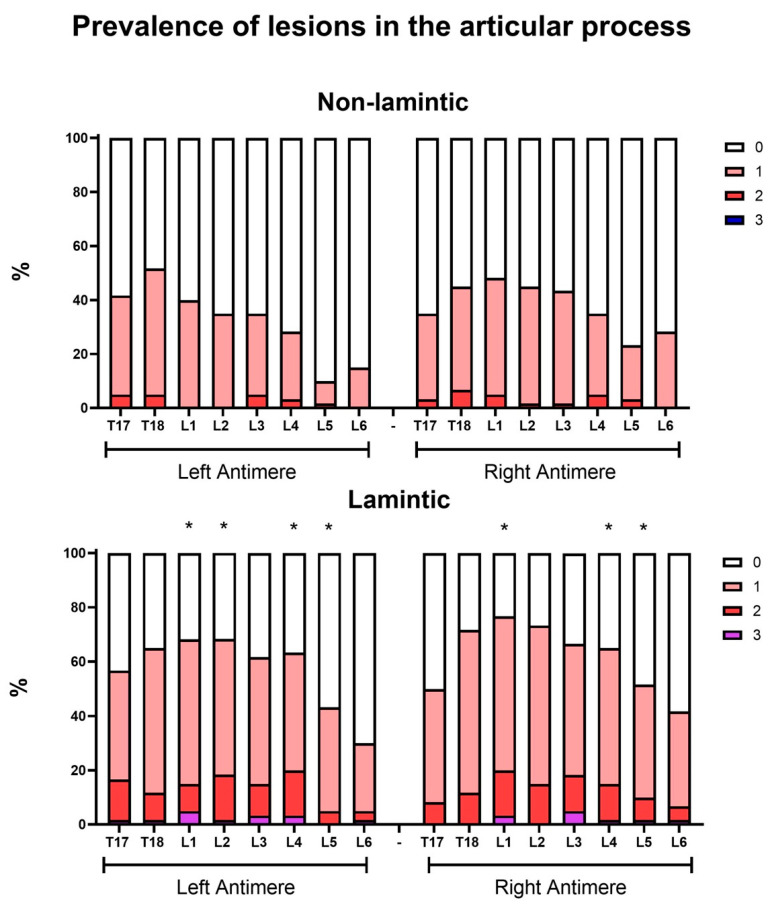
Relative prevalence of injuries in the articular processes (APs) of the thoracolumbar spines of horses with and without chronic laminitis. The asterisks indicate the articular processes in which there was a higher prevalence of injuries (grades 1 to 3) in horses with laminitis compared to the non-laminitic ones (*p* < 0.05).

**Figure 14 animals-14-01364-f014:**
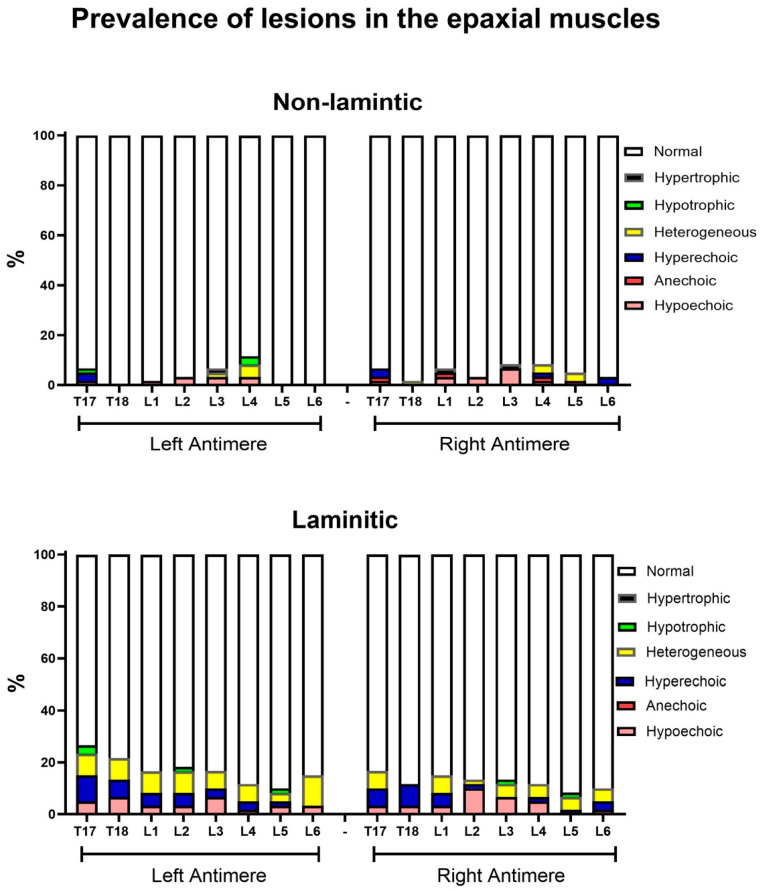
Relative prevalence of ultrasound injuries observed in the epaxial muscle (EM) of horses with or without chronic laminitis.

**Figure 15 animals-14-01364-f015:**
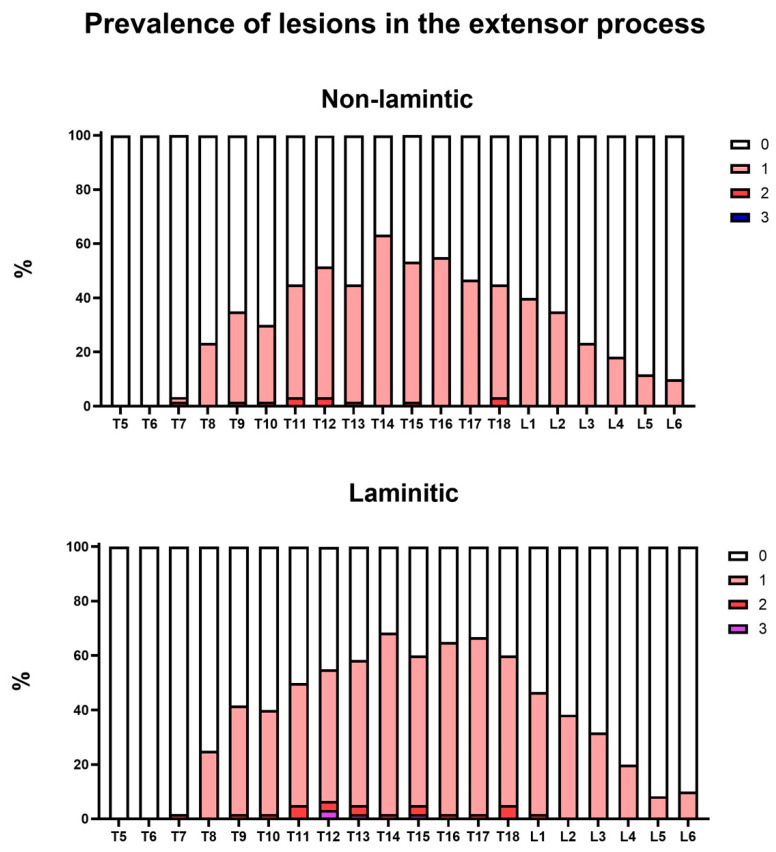
Relative prevalence of injuries in the spinous processes (SPs) of the thoracolumbar spines of horses in the with or without chronic laminitis.

**Figure 16 animals-14-01364-f016:**
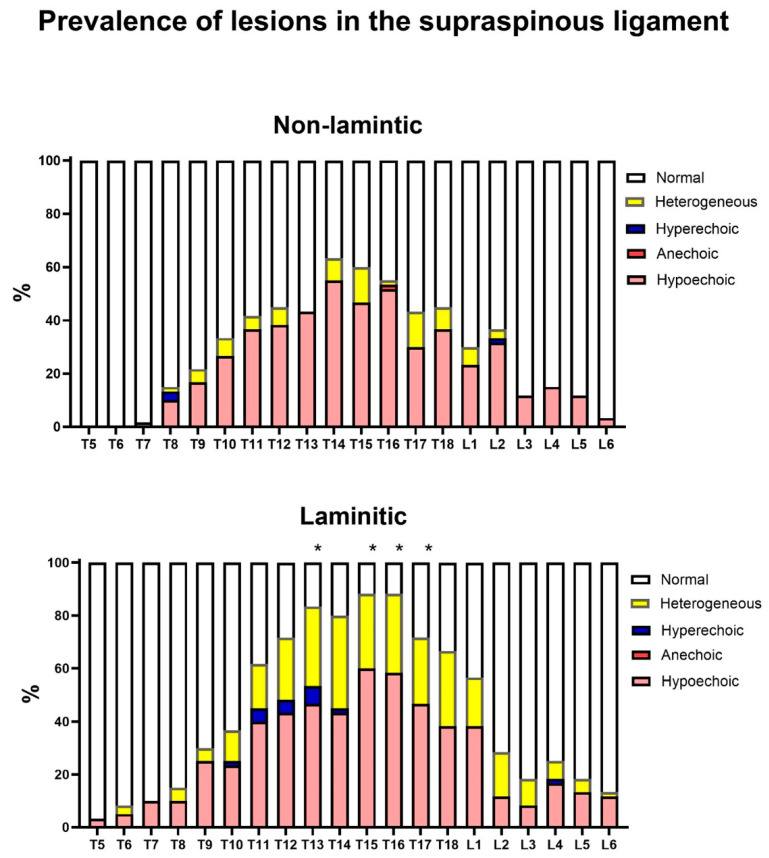
Relative prevalence of injuries in the supraspinous ligament (SSL) of the thoracolumbar spines of horses with and without chronic laminitis. The asterisks indicate the segments of the supraspinous ligament in which there was a higher prevalence of alterations in horses with laminitis compared to the others (*p* < 0.05).

**Figure 17 animals-14-01364-f017:**
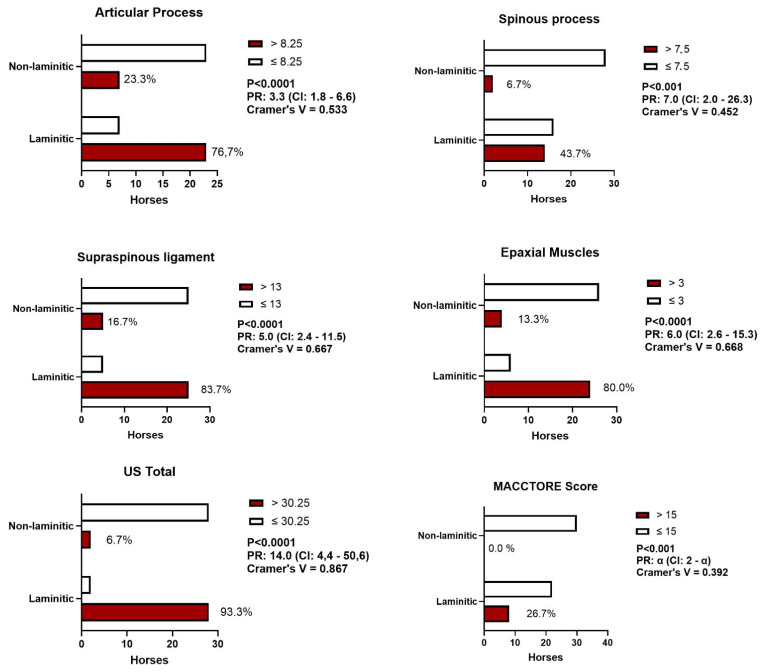
Absolute frequency of horses with and without chronic laminitis with severe injuries (those in the upper quartile—P75) in the thoracolumbar spine considering clinical (MACCTORE) and ultrasound (AP, SP, EM, SSL, and US total) variables.

**Table 1 animals-14-01364-t001:** MACCTORE (Clinical Assessment Method of the Equine Thoracolumbar Spine). The MACCTORE is a tabulation methodology based on a scale of 0–3 and divided into three components: inspection, palpation, and tests of mobility. Within each of these components, there are predefined parameters. The inspection step includes an evaluation of the degrees of muscle atrophy of specific muscle groups, degrees of scoliosis, degrees of lordosis, degrees of kyphosis, degrees of abdominal contraction, and degrees of sacral tuberosity asymmetry. The palpation step includes an assessment of the degree of muscle pain, muscle spasm/tension, and the presence or absence of muscle knots and muscle fasciculation in specific muscle groups; an assessment of pain at the dorsal midline; and an assessment of the misalignment of the spinous processes. Tests of spinal mobility include a graded evaluation of thoracic, thoracolumbar, and lumbosacral extensions, thoracic and lumbosacral flexions, and left and right lateral flexions. For each assessed parameter, a score from 0 to 3 (for inspection and palpation) or 0 to 2 (for mobility tests) was assigned. Each grading within the MACCTORE method has a descriptive definition of how the horse is expected to behave (or not) when corresponding to that score. At the end of the table, there is an area for adding any extra information and observations made during the examination and a list of the muscle groups that are assessed.

Method of Clinical Assessment of the Equine Thoracolumbar Spine (MACCTORE)				
**Inspection**	Grade 0	Grade 1	Grade 2	Grade 3
Muscle atrophy	No clinical evidence	Mild atrophy of muscle groups *	Moderate atrophy of muscle groups *	Severe atrophy of muscle groups *
Scoliosis	No clinical evidence	Mild scoliosis	Moderate scoliosis	Severe scoliosis
Lordosis	No clinical evidence	Mild lordosis	Moderate lordosis	Severe lordosis
Kyphosis	No clinical evidence	Mild kyphosis	Moderate kyphosis	Severe kyphosis
Abdominal contraction	No clinical evidence	Mild abdominal muscle contraction	Moderate abdominal muscle contraction	Severe abdominal muscle contraction
Sacral tuber asymmetry	No clinical evidence	Mild sacral tuber asymmetry	Moderate sacral tuber asymmetry	Severe atrophy of muscle groups *
**Palpation**	Grade 0	Grade 1	Grade 2	Grade 3
Muscle pain	No clinical evidence	Mild reaction to palpation of muscle * and to firm pressure of fingers	Moderate reaction to palpation of muscle * and to firm pressure of fingers	Severe reaction to palpation of muscle * and to firm pressure of fingers
Spasm/muscle tension	No clinical evidence	Presence of mild muscle tension in muscle group * during palpation	Presence of moderate muscle tension in muscle group * during palpation	Presence of severe muscle tension in the muscle group * during palpation
Nodules	No clinical evidence	Presence of nodules in the muscle groups *	-	-
Pain in the dorsal midline	No clinical evidence	Mild reaction to palpation with firm pressure of fingers	Moderate reaction to palpation with firm pressure of fingers	Severe reaction to palpation with firm pressure of fingers
Misalignment of the spinous processes	No clinical evidence	Mild misalignment of the spinous processes	Moderate misalignment of the spinous processes	Severe misalignment of the spinous processes
Muscle fasciculation	No clinical evidence	Mild reaction to muscle group * palpation with mild fasciculation	Moderate reaction to muscle group * palpation with mild fasciculation	Severe reaction to muscle group * palpation with mild fasciculation
**Mobility tests**	Grade 0	Grade 1	Grade 2	Grade 3
Thoracic extension (at T10)	Performs complete movement with no decrease in amplitude. Does not express any resistance to the exam or pain behavior	Performs the movement partially or incompletely. Reduction of amplitude and/or demonstration of pain, e.g., shaking the head or tail, reaction with limbs, and/or vocalization	Refuses examination or tries to escape from it. Reacts, demonstrating pain, e.g., shaking the head or tail, reaction with limbs, and/or vocalization	-
Thoracolumbar extension (at T16)	Performs complete movement with no decrease in amplitude. Does not express any resistance to the exam or pain behavior	Performs the movement partially or incompletely. Reduction of amplitude and/or demonstration of pain, e.g., shaking the head or tail, reaction with limbs, and/or vocalization	Refuses examination or tries to escape from it. Reacts, demonstrating pain, e.g., shaking the head or tail, reaction with limbs, and/or vocalization	-
Lumbosacral extension	Performs complete movement with no decrease in amplitude. Does not express any resistance to the exam or pain behavior	Performs the movement partially or incompletely. Reduction of amplitude and/or demonstration of pain, e.g., shaking the head or tail, reaction with limbs, and/or vocalization	Refuses examination or tries to escape from it. Reacts, demonstrating pain, e.g., shaking the head or tail, reaction with limbs, and/or vocalization	-
Thoracic flexion (at xyphoid cartilage)	Performs complete movement with no decrease in amplitude. Does not express any resistance to the exam or pain behavior	Performs the movement partially or incompletely. Reduction of amplitude and/or demonstration of pain, e.g., shaking the head or tail, reaction with limbs, and/or vocalization	Refuses examination or tries to escape from it. Reacts, demonstrating pain, e.g., shaking the head or tail, reaction with limbs, and/or vocalization	-
Lumbosacral flexion	Performs complete movement with no decrease in amplitude. Does not express any resistance to the exam or pain behavior	Performs the movement partially or incompletely. Reduction of amplitude and/or demonstration of pain, e.g., shaking the head or tail, reaction with limbs, and/or vocalization	Refuses examination or tries to escape from it. Reacts, demonstrating pain, e.g., shaking the head or tail, reaction with limbs, and/or vocalization	-
Left lateral flexion	Performs complete movement with no decrease in amplitude. Does not express any resistance to the exam or pain behavior	Performs the movement partially or incompletely. Reduction of amplitude and/or demonstration of pain, e.g., shaking the head or tail, reaction with limbs, and/or vocalization	Refuses examination or tries to escape from it. Reacts, demonstrating pain, e.g., shaking the head or tail, reaction with limbs, and/or vocalization	-
Right lateral flexion	Performs complete movement with no decrease in amplitude. Does not express any resistance to the exam or pain behavior	Performs the movement partially or incompletely. Reduction of amplitude and/or demonstration of pain, e.g., shaking the head or tail, reaction with limbs, and/or vocalization	Refuses examination or tries to escape from it. Reacts, demonstrating pain, e.g., shaking the head or tail, reaction with limbs, and/or vocalization	-

* Muscle groups that are assessed for atrophy, pain, spasm/tension, presence of nodules, or fasciculations: spinalis; longissimus dorsi; gluteus medius, accessory gluteal muscle, superficial gluteal muscle; quadriceps femoris; semitendinosus and semimembranosus (always identify the muscle portion: origin/insertion/third cranial/mid/third caudal/all extension).

## Data Availability

The data presented in this study are openly available in the repository of Universidade Federal de Minas Gerais (UFMG).

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
