# Peer review of "The Detection of Thoracolumbar Spine Injuries in Horses with Chronic Laminitis Using a Novel Clinical-Assessment Protocol and Ultrasonographic Examination"

_animals, 2024, doi:10.3390/ani14091364_

Round 1

Reviewer 1 Report (Previous Reviewer 2)

Comments and Suggestions for Authors

An interesting study, which will make a contribution to the field and add to discussion re laminitis, however currently the authors are perhaps over emphasising the relationships and impact of their work which should be addressed prior to publication. I would also like to have seen evidence for how the instrument used was developed and how its reliability was assessed included.

Simple summary – appropriate

Abstract – relevant summary, I would suggest adapting the final sentence to reflect your results more, which show an increased presence of spinal injuries in horses which present with laminitis and remove the association as you have not tested for this and we don’t know if these are present due to this or other reasons

Introduction

Line 51: suggest amending to a severe as there are arguably other serious conditions as well

Line 84-89: it would be beneficial to also allude to factors which can increase risk of developing laminitis for example, box rest or periods out of work which could be present with spinal injuries such as kissing spines, therefore there could be a bit of chicken and egg scenario between the back and feet and laminitis, worthy of establishing (and keeping in mind for interpretation)

Materials and methods

Please confirm control group also had no history of back issues

It would be beneficial to also outline ridden / exercise history for horses, if known, as this could impact back muscle profiles and scoring.

Results

Clearly presented

Discussion

Line 436-451: you have identified an interesting potential link but only in a relatively small number of horses and as you allude to without being able to establish causality, so would advise tempering inferences made here a little

Line 444: horses’ history is also worthy of consideration here as management is likely a key factor in development of laminitis

Line 469: parenthesis missing

Line 487: please amend to validated

Line 550-553: as stated above, your results show there is an increased prevalence but do not establish an association and there are many potential confounders and co-factors which could influence this and this needs to be considered here

Conclusions

Again here feel the inferences made and conclusions drawn need to be tempered as you are placing too much emphasis on one study, with a relatively small sample size – you have a really interesting study which has worth for publication, don’t try and make it more than it is; suggest revisiting this section and rewriting

Comments on the Quality of English Language

Couple of minor areas to address 

Author Response

An interesting study, which will make a contribution to the field and add to discussion re laminitis, however currently the authors are perhaps over emphasising the relationships and impact of their work which should be addressed prior to publication. I would also like to have seen evidence for how the instrument used was developed and how its reliability was assessed included.

Simple summary – appropriate

Abstract – relevant summary, I would suggest adapting the final sentence to reflect your results more, which show an increased presence of spinal injuries in horses which present with laminitis and remove the association as you have not tested for this and we don’t know if these are present due to this or other reasons

Thanks for your comments and careful consideration. We made changes in the final sentence to avoid over emphasizing our findings as smartly advised. However, we considered that the word association would be kept. Although this is just preliminary evidence, a statistical association was detected in the present study.

In an epidemiological perspective, association can be defined as a general relationship between two attributes. Two attributes are said to be associated if the response of one attribute changes over the different states/categories of the other attribute. Conversely, if the response does not change, they are said to be independent. In our case, if we notice in the population of horses with chronic laminitis are more prone to develop back pain, we have a reason to suspect that chronic laminitis is linked or associated with occurrence of back pain. Had they been independent, we could have observed a similar response (prevalence) of back pain among laminitic as well as non-laminitic horses.

With this purpose, the experimental design was designed according to directions of the Best Seller and reference book Gordis Epidemiology, 7th Edition, Chapter 7 where it can be read: “Another common study design used in initially investigating the association between a specific exposure and a disease of interest is the cross-sectional study.” In our case, the specific exposure was the consequences of chronic laminitis, and the disease was the thoracolumbar injury/pain. In order to better characterize this study as a cross-sectional design, we removed the control group designation in this reviewed version and used the terms laminitic and non-laminitic horses, which is more appropriate.

Initially, we used the Fisher test followed by the calculation of the prevalence ratio as described by (Tamhane AR, Westfall AO, Burkholder GA, Cutter GR. Prevalence odds ratio versus prevalence ratio: choice comes with consequences. Stat Med. 2016 Dec 30;35(30):5730-5735. doi: 10.1002/sim.7059. Epub 2016 Jul 26). If the calculated P < 0.05, the null hypothesis of independence was rejected and a significant association between the two attributes was detected.

However, in order to double check the association between the attributes and to qualify its magnitude we used another statistical approach. By this time, we compared the prevalence of back pain/lesions between laminitic and non-laminitic horses using the chi-square test. Furthermore, the frequency distribution of horses affected with scores of clinical and severe US injuries, whose cutoff point corresponded to the 75th percentile (P75) of the entire population in each variable were equally compared. If the calculated P < 0.05, the null hypothesis of independence was rejected and a significant association between the two attributes was detected. The strength and relevance of statistical significant associations between exposure and specific outcomes parameters were subsequently evaluated trough the calculation of the Cramer's V coefficient (Sapra, Ramesh Lal1,*; Saluja, Satish2. Under-standing statistical association and correlation. Current Medicine Research and Practice 11(1):p 31-38, Jan–Feb 2021. | DOI: 10.4103/cmrp.cmrp_62_20)  and the calculation of the prevalence ratio (Martinez BAF, Leotti VB, Silva GS, Nunes LN, Machado G and Corbellini LG (2017) Odds Ratio or Prevalence Ratio? An Overview of Reported Statistical Methods and Appropriateness of Interpretations in Cross-sectional Studies with Dichotomous Outcomes in Veterinary Medicine. Front. Vet. Sci. 4:193. doi: 10.3389/fvets.2017.00193). The Cramer’s V coefficient (scale 0 to 1) was obtained as previously described (Sapra & Saluka), and the association between the exposure factor and outcome was classified as negligible (up to 0.10), weak (0.11 to 0.20), moderate (0.21 to 0.40), relatively strong (0.41 to 0.60), strong (0.61 to 0.80), and very strong (0.81 to 1.00).

This information was included in the itens 2.1 and 2.5 of the Material and Methods Section.

Introduction

Line 51: suggest amending to a severe as there are arguably other serious conditions as well

OK

Line 84-89: it would be beneficial to also allude to factors which can increase risk of developing laminitis for example, box rest or periods out of work which could be present with spinal injuries such as kissing spines, therefore there could be a bit of chicken and egg scenario between the back and feet and laminitis, worthy of establishing (and keeping in mind for interpretation)

Yes, this is an interesting point. It was included in the first paragraph in the discussion section.

Materials and methods

Please confirm control group also had no history of back issues

OK, done

It would be beneficial to also outline ridden / exercise history for horses, if known, as this could impact back muscle profiles and scoring.

All the available information is presented in the supplementary material. We changed one of the tables to better characterize the ridden/exercise history in each group.

Results

Clearly presented

Discussion

Line 436-451: you have identified an interesting potential link but only in a relatively small number of horses and as you allude to without being able to establish causality, so would advise tempering inferences made here a little

OK, the paragraph was changed, and this information was further discussed again in the limitations paragraph.

Line 444: horses’ history is also worthy of consideration here as management is likely a key factor in development of laminitis

Ok, included.

Line 469: parenthesis missing

OK

Line 487: please amend to validated

OK

Line 550-553: as stated above, your results show there is an increased prevalence but do not establish an association and there are many potential confounders and co-factors which could influence this and this needs to be considered here

We included new information and used a new statistical approach to confirm and even estimate the strength of the association between chronic laminitis and thoracolumbar lesion. The need of more sophisticated studies to control confounders and co-factors we included in the limitations paragraph.

Conclusions

Again here feel the inferences made and conclusions drawn need to be tempered as you are placing too much emphasis on one study, with a relatively small sample size – you have a really interesting study which has worth for publication, don’t try and make it more than it is; suggest revisiting this section and rewriting

Great, the conclusions were rewritten.

Thanks again!

Reviewer 2 Report (New Reviewer)

Comments and Suggestions for Authors

The authors provided a thorough manuscript that provides the results of a fairly-labor intense study.  This work provides evidence that there is a link between horses that have experienced laminitis and subsequent spinal injuries.  The study design has some limitations, the majority of which are discussed by the authors.  One item that I did not see addressed but feel should is that it appears (unless I missed it) that even though the same person was responsible for grading the various items, that person was not blinded to whether the horses were in the control group or the laminitic group.  Particularly with scoring being somewhat subjective, this is a limitation to the study that should be addressed.  (As a researcher, I understand how often the person in charge of the study also has to take measurements, thus making blinding challenging, but it does bring about the potential for bias.)  Further, though training was done to ensure the person making the measurements knew what they were doing, there was no mention of any attempts to determine repeatability of the scoring.  This is another limitation and should be addressed.  Overall, the paper was reasonably well-written and it was clear the authors put in great effort into both conducting the study and the preparation of the manuscript.

Lines

107-109  In the appendix, some details are given as to breeds and other demographics of the horses in Table A2.  However, it would be more useful if all the information was completely separated out by group (CON vs CLG) as was done in Table A3.

110  Change to “came from different breeding farms…”

124  Remove comma after Spine

170, 174 (as well as a few other lines in the manuscript such as 479)  When the number of the citation is given, it is a bit redundant to give the year of the publication also.  

274  Remove comma after assessment

275            There is a wording issue with this sentence (“were blindly assessed other two”).  I assume you mean “were blindly assessed by two other veterinarians”.

358-359  The graph seems to indicate that the control horses had a higher RT in contrast to what the text indicates

487  Change to “validated by”

523  Change to “worth considering”

530  Change to “is further evidence”

Please double check all citations to make sure they are formatted correctly.

Comments on the Quality of English Language

The effort the authors took in preparation of the manuscript was clear and there were only minor recommendations for wording changes.

Author Response

Reviwer 2

The authors provided a thorough manuscript that provides the results of a fairly-labor intense study.  This work provides evidence that there is a link between horses that have experienced laminitis and subsequent spinal injuries.  The study design has some limitations, the majority of which are discussed by the authors.  One item that I did not see addressed but feel should is that it appears (unless I missed it) that even though the same person was responsible for grading the various items, that person was not blinded to whether the horses were in the control group or the laminitic group.  Particularly with scoring being somewhat subjective, this is a limitation to the study that should be addressed.  (As a researcher, I understand how often the person in charge of the study also has to take measurements, thus making blinding challenging, but it does bring about the potential for bias.)  Further, though training was done to ensure the person making the measurements knew what they were doing, there was no mention of any attempts to determine repeatability of the scoring.  This is another limitation and should be addressed.  Overall, the paper was reasonably well-written and it was clear the authors put in great effort into both conducting the study and the preparation of the manuscript.

Thanks for recognizing the efforts of the authors conducting this work and for your careful review and relevant considerations.

Yes, the MACCOTRE was scored always by the same researcher that was aware of the laminitis status of the horse due to the characteristic posture of the affected horses. However, the US images evaluation was conducted by two independent and blind researchers. Limitations about a single group and new methods that were used were further stressed in the discussion section.

Lines

107-109  In the appendix, some details are given as to breeds and other demographics of the horses in Table A2.  However, it would be more useful if all the information was completely separated out by group (CON vs CLG) as was done in Table A3.

OK

110  Change to “came from different breeding farms…”

OK

124  Remove comma after Spine

OK

170, 174 (as well as a few other lines in the manuscript such as 479)  When the number of the citation is given, it is a bit redundant to give the year of the publication also.  

OK

274  Remove comma after assessment

OK

275            There is a wording issue with this sentence (“were blindly assessed other two”).  I assume you mean “were blindly assessed by two other veterinarians”.

OK, written.

358-359  The graph seems to indicate that the control horses had a higher RT in contrast to what the text indicates

Yes, thanks!

487  Change to “validated by”

OK

523  Change to “worth considering”

OK

530  Change to “is further evidence”

OK

Please double check all citations to make sure they are formatted correctly.

OK, many thanks!

This manuscript is a resubmission of an earlier submission. The following is a list of the peer review reports and author responses from that submission.

Round 1

Reviewer 1 Report

Comments and Suggestions for Authors

This study aims to investigate thoracolumbar equine spine injuries and chronic laminitis. The title suggests that some sort of association will be evaluated between the two clinical conditions, however, this is not the supported by the manuscript.

The Authors state:  ’proposal raised by this research is to experimentally investigate if such a relationship exists and to what extent this change in posture adopted by the horse during chronic laminitis reflects on the thoracolumbar spine and then, what are the main injuries that can occur as a result.’

These aims are already overambitious and such relationship could only be investigated in a more controlled study, where data are collected or information on the horses’ thoracolumbar spine is available prior to developing lamintis. This is very unlikely to be possible in a clinical setting with naturally occurring laminitis.

It would be crucial to provide much more detailed information on case selection and the clinical symptoms shown by laminitic horses. Some chronic laminitis cases can show minimal to no lameness and would therefore not be expected to alter their posture and develop thoracolumbar abnormalities.

The presence of thoracolumbar abnormalities in the laminitic group is by no means a proof of causal relationship. All of the investigated clinical and imaging findings can be associated with many other primary or secondary causes. In a future study, with much clearer and robust case selection (e.g., defining the level of pain or degree of lameness caused by laminitis), more robust statistical analysis should be performed controlling for other risk factors and potential confounders.

The paper does not follow the journal’s guidelines. The Results should be presented without comments and explanations, and a separate Discussion should follow. The number of references is low and not the most suitable or relevant sources are cited in many parts of the paper.

In summary, in my opinion this manuscript is not suitable for publication and the flaws in the study design are too big to allow revision. If Authors wish to investigate their research question further, they have to select their cases carefully and in their analysis and interpretation consider all factors that can contribute to clinical and imaging abnormalities of the thoracolumbar spine.   

Author Response

Thanks for your careful evaluation and all your valuable comments. It’s important to note that this is an epidemiological study based only in observation instead of an experimental study where we are able to control and manipulate the conditions and exposure for individuals.

We agree that our results would be more powerful if we did a longitudinal cohort study, when the group of subjects comprising the cohort is chosen based on a disease or injury and then followed at routine intervals over time. Specially, if we had the opportunity to start to examine the horses back even before they suffer from laminitis, as you have mentioned. However, we considered that this scenario would be quite unrealistic, since we had to serially exam the back of an enormous population kept in the same conditions of nutrition, management, and training/working for a long period of time, waiting for few of them to luckily (or unluckily) develop abnormal posture due to laminitis.

So, we opted to a cross-sectional study, which makes observations of a representative subset of a population at a specific point in time. This design is often used to assess a population sample to see if a previous condition or risk factor, say chronic laminitis, is related to the health effect being investigated, say back pain and lesions. We are aware that this type of design cannot be used to determine the causes of disease because temporality is not known. However, the associationbetween the previous condition (abnormal posture due to laminitis) and the studied health effect (back pain and lesions) can be established and measured using traditional statistical methods. (Beichou, J., Palta, M. (2014). Rates, Risks, Measures of Association and Impact. In: Ahrens, W., Pigeot, I. (eds) Handbook of Epidemiology. Springer, New York, NY. https://doi.org/10.1007/978-0-387-09834-0_3)

We recognize that the reported associations are only initial evidence, and that new and more complex studies are needed to better understand the causality and relevance of this relationship. We made several changes in the manuscript to avoid overambitious aims and conclusions, and to better explain the study limitations.  

Reviewer 2 Report

Comments and Suggestions for Authors

General comment:  

Interesting study that is adding data to the debate between the interrelationship between equine back pain and feet issues. Which came 1st needs to be considered and addressed more across the manuscript.

Would be beneficial to include some more discussion about diseases related to laminitis such as EMS and PPID and potential for results to apply to these groups also

Simple summary: clear summary of study, would be beneficial to add in no of horses used here

Abstract: concise summary, would be beneficial to add outline of what statistical approaches have been used here for clarity

Introduction

Line 80: as anecdotal suggest adding anecdotally it is reported that horses … to set into context

It would be beneficial to expand on the detail of what a laminitic posture is and why it occurs including describing the biomechanical impact for readers less familiar with the area

Would also suggest linking laminitis to EMS and PPID in your introduction

Materials and methods

Line 104/105: comment in brackets need addressing

Line 107: edit sentence as this doesn’t currently make sense

Line 1134: please explain Obel gradings

Line 114: assume the control horses also had no prior history of laminitis – please can you add info to clarify

Line 128: it is pleasing to see new tools being developed, but there does need to be some details here to demonstrate the reliability and validity of this approach and how this has been tested before undertaking this study and increase detail of what evidence sources were utilised to underpin this

Are there any specific requirements for how the test is undertaken? Please detail

Please also outline details of the training undertaken and the experience of the researcher used in the study

Generally each phase of the testing is well described but feel scope to increase justification through reference to underpinning literature

Given the dynamic relationship between the horse’s back and feet, and potential that some spinal issues could have been present prior to laminitis symptoms, I feel this needs to be addressed either here or in a limitations section and potentially wider in the discussion. Generally pre-existig issue sor prior management needs to be considered more here as the control and treatment horses have not been matched therefore, muscle profiles for example could be the result of cumulative impact of ongoing spinal or feet related, poor training etc and not necessarily be associated with the onset of acute or chronic laminitis (although they may have contributed to problems).

There is also no details of the medication and treatment protocols horses were on when testing occurred e.g. analgesic and antiflammatories – this information should be provided

Results and discussion

Line 366: I would urge caution making this statement as strongly given weight is a key risk factor to laminitis

Line 374: ref needs to be inserted were noted

Given nature of study, suggest editing to present as results and separate discussion

The results would benefit from a judicious edit to make more concise and to the point and improve synthesis in sentence construction and enhance impact.

A specific limitations section should be included – also see previous comments regarding areas to integrate into discussion and also include a section which considers the new tool and its reliability and validity

Conclusions

Line 601: please amend proving to suggesting – this is 1 preliminary study which does have some substantial limitations, but does add some interesting data into this field

Line 606-608: for me this sentence is the key discussion point you have, rather than try to identify and suggest a causal relationship here, for me this more around a dynamic interrelationship and we don’t know enough yet to know how these interact as risk factors from your work, byut there is more likelihood that laminitic horses have these present and therefore further work to understand this relationship is needed and that would be your key take home message

Final paragraph should be in discussion not conclusion

References – please ensure these align to journal requirements

Missing conflict of interest, acknowledgement etc statements - please refer to journal requirements

Comments on the Quality of English Language

An edit to increase impact and synthesis is advised

Author Response

Comments and Suggestions for Authors

General comment: 

Interesting study that is adding data to the debate between the interrelationship between equine back pain and feet issues. Which came 1st needs to be considered and addressed more across the manuscript.

OK, done.

Would be beneficial to include some more discussion about diseases related to laminitis such as EMS and PPID and potential for results to apply to these groups also

OK, included.

Simple summary: clear summary of study, would be beneficial to add in no of horses used here

OK, done.

Abstract: concise summary, would be beneficial to add outline of what statistical approaches have been used here for clarity

Introduction

Line 80: as anecdotal suggest adding anecdotally it is reported that horses … to set into context

It would be beneficial to expand on the detail of what a laminitic posture is and why it occurs including describing the biomechanical impact for readers less familiar with the area.

Lines 63-72

Would also suggest linking laminitis to EMS and PPID in your introduction

OK, done.

Materials and methods

Line 104/105: comment in brackets need addressing

OK, done.

Line 107: edit sentence as this doesn’t currently make sense

OK, thanks.

Line 1134: please explain Obel gradings

OK, done.

Line 114: assume the control horses also had no prior history of laminitis – please can you add info to clarify

OK, included.

Line 128: it is pleasing to see new tools being developed, but there does need to be some details here to demonstrate the reliability and validity of this approach and how this has been tested before undertaking this study and increase detail of what evidence sources were utilised to underpin this

Are there any specific requirements for how the test is undertaken? Please detail

Please also outline details of the training undertaken and the experience of the researcher used in the study

Generally each phase of the testing is well described but feel scope to increase justification through reference to underpinning literature

OK, thanks for your comments! More and detailed information was provided. (Lines 129-138).

Given the dynamic relationship between the horse’s back and feet, and potential that some spinal issues could have been present prior to laminitis symptoms, I feel this needs to be addressed either here or in a limitations section and potentially wider in the discussion. Generally pre-existig issue sor prior management needs to be considered more here as the control and treatment horses have not been matched therefore, muscle profiles for example could be the result of cumulative impact of ongoing spinal or feet related, poor training etc and not necessarily be associated with the onset of acute or chronic laminitis (although they may have contributed to problems).

There is also no details of the medication and treatment protocols horses were on when testing occurred e.g. analgesic and antiflammatories – this information should be provided

OK, a paragraph about such limitations was included in the discussion section.

Results and discussion

Line 366: I would urge caution making this statement as strongly given weight is a key risk factor to laminitis

OK, statement was rewritten.

Line 374: ref needs to be inserted were noted

Given nature of study, suggest editing to present as results and separate discussion

The results would benefit from a judicious edit to make more concise and to the point and improve synthesis in sentence construction and enhance impact.

A specific limitations section should be included – also see previous comments regarding areas to integrate into discussion and also include a section which considers the new tool and its reliability and validity

OK, done!

Conclusions

Line 601: please amend proving to suggesting – this is 1 preliminary study which does have some substantial limitations, but does add some interesting data into this field

Line 606-608: for me this sentence is the key discussion point you have, rather than try to identify and suggest a causal relationship here, for me this more around a dynamic interrelationship and we don’t know enough yet to know how these interact as risk factors from your work, byut there is more likelihood that laminitic horses have these present and therefore further work to understand this relationship is needed and that would be your key take home message

Final paragraph should be in discussion not conclusion

OK, done! Thanks for your great comments!

References – please ensure these align to journal requirements

Missing conflict of interest, acknowledgement etc statements - please refer to journal requirements

Comments on the Quality of English Language

An edit to increase impact and synthesis is advised

Done!

Round 2

Reviewer 1 Report

Comments and Suggestions for Authors

The Authors made extensive revision and parts of the manuscript have improved significantly. Unfortunately, it does not change the major flaw: comparison of thoracolumbar pain and lesions in two groups of horses with minimal clinical information does not yield reliable results.

If Authors wish to resubmit the manuscript, I would advise putting much more focus on clinical aspects and summarising the clinical and imaging findings in the TL region in fewer tables.

Authors would also need to support the background of the study. Authors state in the Introduction: ‘Even when the horse recovers from laminitis and begins to improve, it often does not return to a normal posture, or still has difficulties with locomotion and/or an inability to resume an athletic life.’ This sentence would need to be supported by evidence-based publications. There may also be some confusion about terminology – if the horse has recovered from lamintis, why would it have altered posture or lameness? Those suggest that the horse has not recovered…

While there are very detailed descriptions of clinical and imaging findings in the thoracolumbar region, only minimal information of the horses is provided. Authors put a lot of emphasis on association between clinical and imaging scores of TL findings, which is interesting, but not the main focus of the study.

The following would be much more important in my view, and they would be crucial for accurate interpretation of the results:

-       Potential effect of age (there is only general description pooling the two groups)

-       Why did Authors not attempt to use age and breed matched controls?

-       What clinical signs did the horse show at the time of examination? Posture, gait etc.

-       How long prior to the examination was the horse diagnosed with laminitis? More specific information than 3-12 months would be needed.

-       Which limb(s) were involved?

-       How severely were the horses affected by laminitis? Was there an association between the clinical/radiological signs and TL pain or lesions?

-       Was there an association between the affected limb(s) and the location and severity of T/L lesions?

Author Response

The Authors made extensive revision and parts of the manuscript have improved significantly. Unfortunately, it does not change the major flaw: comparison of thoracolumbar pain and lesions in two groups of horses with minimal clinical information does not yield reliable results.

Thanks for valuable consideration! Studying the first of round of reviews we identify that the real objective of this work was not enough clear. However, as noted, we have made improvements in our hypothesis and objectives. Please refer to the lines 96 to 100. “…the research delves into the hypothesis positing an association between thoracolumbar spine lesions and laminitis in horses. Employing a cross-sectional approach, the study aimed to identify and compare the prevalence rates of these lesions in both clinically healthy and laminitis-affected horses. Furthermore, the investigation assessed the magnitude of these lesions and compare their associated clinical signs between the two groups.

Please note that at no point was it stated that the goal of this study was to examine the previous history or any clinical data of laminitic horses and attempt to establish a direct relationship between any semiological detail and specific thoracolumbar pain or lesions found during the current examination. Such a goal is beyond the scope of our methods.

If Authors wish to resubmit the manuscript, I would advise putting much more focus on clinical aspects and summarising the clinical and imaging findings in the TL region in fewer tables.

Thanks for advice. We changed the title to fit it better to our objectives and included more information about the limitations of our methodology in the discussion (lines 568-570). We tried to summarize the graphic representation of our results, but unfortunately the generated figure wasn’t clear enough to show them.

Authors would also need to support the background of the study. Authors state in the Introduction: ‘Even when the horse recovers from laminitis and begins to improve, it often does not return to a normal posture, or still has difficulties with locomotion and/or an inability to resume an athletic life.’ This sentence would need to be supported by evidence-based publications. There may also be some confusion about terminology – if the horse has recovered from lamintis, why would it have altered posture or lameness? Those suggest that the horse has not recovered…

The text was changed to improve clarity and evidence-based reference added.

While there are very detailed descriptions of clinical and imaging findings in the thoracolumbar region, only minimal information of the horses is provided. Authors put a lot of emphasis on association between clinical and imaging scores of TL findings, which is interesting, but not the main focus of the study.

OK, we made new changes to explain that our main focus was solely to find an statistical association between clinical and imaging scores of TL.

The following would be much more important in my view, and they would be crucial for accurate interpretation of the results:

-       Potential effect of age (there is only general description pooling the two groups)

-       Why did Authors not attempt to use age and breed matched controls?

-       What clinical signs did the horse show at the time of examination? Posture, gait etc.

-       How long prior to the examination was the horse diagnosed with laminitis? More specific information than 3-12 months would be needed.

-       Which limb(s) were involved?

-       How severely were the horses affected by laminitis? Was there an association between the clinical/radiological signs and TL pain or lesions?

-       Was there an association between the affected limb(s) and the location and severity of T/L lesions?

Yes, we completely acknowledge the importance of these aspects in comprehending the causal effect of potential risk factors. However, our study was solely designed to test the hypothesis of the existence of this association. Certainly, future studies could benefit from more robust designs, such as cohort models with age and breed matched controls, along with comprehensive data collection including details of laminitis history, clinical and radiographic assessment of the foot, and other pertinent parameters potentially linked to specific thoracolumbar lesions. We hope that this initial study serves as a catalyst to stimulate other research groups to get involved in this field.